# The Emotions, Coping, and Psychological Well-Being in Time of COVID-19: Case of Master’s Students

**DOI:** 10.3390/ijerph19106014

**Published:** 2022-05-15

**Authors:** Audrone Dumciene, Jurate Pozeriene

**Affiliations:** 1Department of Physical and Social Education, Lithuanian Sports University, Sporto 6, LT-44221 Kaunas, Lithuania; 2Department of Health Promotion and Rehabilitation, Lithuanian Sports University, Sporto 6, LT-44221 Kaunas, Lithuania; jurate.pozeriene@lsu.lt

**Keywords:** master’s students, psychological well-being, emotions, coping, COVID-19

## Abstract

Background: Master’s students have been affected by COVID-19 and the changing study conditions due to the lockdown. The aim was to uncover changes in emotions, coping strategies, and psychological well-being during a pandemic. Methods: Ryff scale, multidimensional emotion questionnaire, and Brief COPE scale. Participants: sample of 118 master’s students after the first wave and 128 master’s students after the second wave. Results: After the second wave of COVID-19, the happy, enthusiastic, and inspired scores of the emotion construct components increased statistically significantly (*p* < 0.05), but the scores of the components sad, afraid, angry, ashamed, and anxious decreased significantly (*p* < 0.05). After the first wave, students commonly used planning, positive reframing, self-blame, humor, and acceptance coping strategies, which are classified as problem-focused and emotion-focused coping strategies. The psychological well-being of master’s students after the second wave was statistically (*p* < 0.05) better than that after the first wave in many indicators. Environment mastery skills did not change significantly. Significant associations were revealed between the same components of psychological well-being, emotion, and coping strategies. Conclusions: This study showed that the master’s students improved their adaptive abilities probably in the environment of long-term exposure to coronavirus disease, as most psychological well-being indicators improved significantly after the second wave.

## 1. Introduction

There are many different concepts of well-being, psychological well-being, emotions, and coping in the scientific literature. We use some of the concepts in this study. Well-being is defined as “a dynamic optimal state of psychosocial functioning that arises from functioning well across multiple psychosocial domains” [1] (p. 2). The psychological well-being concept includes “self-acceptance, positive relations with others, autonomy, environmental mastery, purpose in life, and personal growth” [2] (p. 1071).

“Emotion is a complex set of interactions among subjective and objective factors, mediated by neural/hormonal systems” [3] (p. 355). The emotion construct in this study is as described by the questionnaire developers [4]. The construct of emotions includes both discrete emotions and generalizing dimensions that include changes in emotional experience over time, emotional reactivity, and the regulation of emotions.

Coping can be defined as “the cognitions and behaviors, adopted by the individual following the recognition of a stressful encounter, that are in some way designed to deal with that encounter or its consequences” [5] (p. 7). The coping construct can be defined as the entire psychological and physiological processes that take place in a person through willpower and/or involuntary processes in a stressful environment.

The spread of the coronavirus disease (COVID-19) has changed the lives of many people around the world, including students, and they used various coping strategies. Distance learning, limited social contacts, and quarantine have changed many of their activities and lifestyle habits, and have had an impact on their health and psychological well-being. Psychological well-being is associated with the harmony of life, and it is “not a stable derivative, and although it is formed in man as an adult, it changes over time” [6] (p. 218). The COVID-19 pandemic affected the mental health of social science students, causing severe emotional exhaustion and deteriorating mental health, and causing stress, irritability, anxiety, and depressive symptoms [7].

The effects of a COVID-19 pandemic can change people’s emotional state and psychological well-being due to isolation, distance learning, and other factors. The results of the study showed that the strategies for regulating emotions differ according to gender and competencies in the impact of a pandemic [8]. However, it can also be influenced by individual emotional regulation abilities, which also affect the subjective psychological well-being of life [9].

The mental health and well-being of adults were affected during the COVID-19 pandemic, and the mental health outcomes of young people (18–29 years) were even worse [10]. The study has highlighted that the mental health and well-being of adults in the UK were already affected in the first phase of the pandemic [10]. The study found that feelings of happiness and life satisfaction among subjects decreased by 12% as a result of the pandemic [11].

People have experienced the greatest negative impact on their relationships with others. Younger people were more affected. However, those who were physically active before the pandemic had less of an impact. During COVID-19, the vast majority of subjects felt the adverse effects of the pandemic on their well-being [12].

It was revealed that some students began to use more alcohol and drugs during the pandemic, with most deteriorating in mental health [13]. During the COVID-19 pandemic, the level of mental health problems and suicide risk among students also increased significantly and the level increased with increasing periods of isolation [14].

Anxiety, depression, and alcohol abuse increased, and the overall quality of life among students deteriorated during the COVID-19 period, similar to that in society [15]. For example, anxiety rates among Roma university students increased significantly during the pandemic, due to the inability to attend university and meet friends and partners, which increased the risk of distress for university students and may have long-term effects on mental health. Following such stressful situations, it is considered appropriate to provide long-term psychological counseling services to students to increase resources for coping with stress [16].

It was found that students who felt high levels of psychological well-being were more likely to choose active coping styles, while those with lower levels of psychological well-being were more likely to choose avoidance coping strategies. It has been suggested that universities should develop programs to explicate students’ ability to choose active stress management strategies [17].

The results of studies revealed that the well-being during COVID-19 was significantly worse than before the onset of the pandemic. The results of the study demonstrated the importance of coping strategies in overcoming stress in extreme situations. During the pandemic, self-efficacy, optimism, hope, and resiliency were significant predictors of well-being [18]. Active coping strategies correlate significantly with psychological well-being [19]. Thus, research shows that the impact of a pandemic on student status indicators varies.

It was also found that students’ fear of COVID-19 was below average, their psychological well-being was above average, and their life satisfaction was below average [20], and this confirms the results of the study [18]. The impact of the COVID-19 pandemic on student alcohol use and its association with changes in psychological health was investigated. Alcohol consumption was found to have risen sharply during the pandemic, resulting in a deterioration in psychological health. However, psychological health has also been influenced by other factors such as financial resources, social and academic environment, and deteriorating time management skills [21]. The form of moderate depression was revealed in 28% of students during the pandemic. However, a deep form of depression was not observed among the students surveyed [22].

Three groups of factors biologically, psychologically, and socio-economically significantly influence psychological well-being. Different coping strategies were observed in groups of different psychological well-being levels. More intense coping strategies were observed in the higher psychological well-being group and lower-intensity coping strategies in the lower psychological well-being group. Coping for an individual’s physical and mental health, and social connections are critical to the sustainable reduction in a pandemic’s impact [23].

As mentioned earlier, the coping strategies of people and their psychological well-being are influenced by many different factors. It was observed from the results of a study in small U.S. towns that their populations are likely to be psychologically better at coping with the stress of a pandemic than those in large cities [24]. Perceived social support containing physical activity and awareness that you are well-informed about the virus, and ways to protect yourself may be significant prognostic factors for psychological well-being in pandemic isolation, but medical perceptions and movement restrictions were not significant for a psychological well-being prognosis [25]. A study conducted among nursing students found that the coping level of this group of students is significantly higher than the overall level in the country, it is influenced by many factors, and it is likely to be of great importance to public health for them [26]. Examination of the university student’s psychological well-being identified different levels (low, medium, and high levels) of depression, anxiety, and stress. Signs of the influence of the pandemic were weaker for those who had the support of family and relatives [27]. To cope with the impact of the pandemic environment, students more often chose problem-focused and emotions-focused coping stress strategies compared to avoidance coping strategies. Medical students, among other students, best coped with emotional distress, for whom this is likely to be very important in their future careers [28].

A survey of students conducted at the peak of the COVID-19 pandemic found that agreeable individuals tend to follow government rules and recommendations to combat COVID-19, while emotionally less stable individuals tend to stockpile, feel insecure, and fear exposure. The use of various precautions to limit contact with other people often causes negative emotions. Restricting contact beyond the immediate family reveals a higher level of negative emotions [29]. Higher body mass index and poorer self-esteem also harm the mental health of students [30].

Among college students during the pandemic period, physical activity and resilience were negatively correlated with negative emotions, and resilience was positively correlated with physical activity. The results of the study showed that physical activity not only weakens negative emotions but improves the resilience and mental health of college students [31].

Distance learning for students did not have a significant effect on emotion regulation, which was found in a study of first-year medical students before and after COVID-19. Students reported experiencing positive rather than negative emotions more often [32].

A study of positive and negative emotions during the pandemic period found that students use different coping mechanisms and manifest themselves in a variety of positive and negative emotion-mediating roles [33]. A study of students’ links to physical activity before and after the pandemic revealed a variety of such links. Some indicated an intuitive sense of increased need for physical activity, while others reported a decreased motivation for physical activity. The first ones also felt the need to take care of themselves and choose positive coping strategies [34]. It was found that students mostly used, during the COVID-19 pandemic, varieties of coping strategies such as acceptance, planning, and seeking emotional support, and their choices were influenced by gender, age, and place of residence. The coping skills of the youngest students were worst [35].

It was established that during the pandemic, the most commonly used coping strategies for students were acceptance, active coping, and physical activity. Postgraduate students used more coping strategies than bachelor’s students. Coping strategies were significantly correlated with psychological well-being [36]. Most undergraduates during the pandemic experienced symptoms of depression, anxiety, and stress. These indicators were significantly correlated with life satisfaction, psychological well-being, and adaptive coping. Symptoms were more pronounced in young students aged 18–24 years compared to older ones [37]. The individuals in a COVID-19 pandemic environment were affected by a variety of stressors that cause a variety of emotions, challenge a variety of coping strategies, and affect a person’s psychological well-being, as revealed by an analytical review of the results of studies conducted in recent years [7,8,9,10,11,12,13,14,15,16,17,18,19,20,21,22,23,24,25,26,27,28,29,30,31,32,33,34,35,36,37]. As most of the studies mentioned above were more focused on undergraduate students, it is important to investigate how the pandemic environment affected students’ emotions, coping strategies, and psychological well-being among master’s students. Master’s students are a specific part of the students with more study experience, more social connections, and more social commitment.

The study aimed to reveal the expression of emotions, coping strategies, and psychological well-being of master’s students during the two waves of the COVID-19 pandemic period and the associations of emotion, coping, and psychological well-being.

Hypothesis—the students’ coping experience, during the coronavirus disease period, will increase, their psychological well-being will improve, they will feel more positive emotions, and they will more often use problem-focused coping strategies after the second wave compared to the state after the first wave of COVID-19.

## 2. Materials and Methods

### 2.1. Participants and Procedures

Participants were selected by purposive sampling from master’s study programs in the area of social sciences. The Master’s students are a specific part of the student community with more study experience, more social connections, and more social commitment. The sample consisted of 118 full-time master’s students (67 females and 51 males) after the first wave of COVID-19 and 128 master’s students (74 females and 54 males) after the second wave from universities in Lithuania. Subjects in the first and second surveys were selected from the same universities, all Caucasian type. Everyone participated in the study voluntarily, with no financial incentive, and they were informed of their right to terminate their participation in this investigation at any time.

The first wave of coronavirus disease (lockdown) in Lithuania was from 16 March 2020 to 17 June 2020. The second wave (lockdown) was from 7 November 2020 to 30 June 2021. The first survey was conducted from 20 June 2020 until 15 July 2020. The second survey was conducted from 1 July 2021 to 15 July 2021.

The research was conducted following the principles of reliability, honesty, respect, and accountability. The Ethics Committee of Social Sciences Research of the Lithuanian Sports University issued a permit to conduct this research as meeting the ethical and legal requirements in Lithuania, where the research was conducted. The researchers provided participants with information about the study, its goals and objectives, and the progress of the study. Subjects were informed that their data would be processed and stored following the requirements of the Personal Data Protection Code. Subjects were provided with questionnaires, which they completed during the sessions, and the duration of the process was not limited. Subjects were able to express their agreement or refusal to participate in the study by completing the questionnaire and marking one of the possible answers at the beginning of the questionnaire in the sociodemographic part of the questionnaire “I agree to participate” or “I disagree to participate”. Data were collected after the first and second COVID-19 wave lockdowns were canceled.

### 2.2. Methods

The study used the three following scales and questionnaire: the Ryff Scale for Measuring Psychological Well-Being [2], the Multidimensional Emotion Questionnaire [4], and the Brief COPE scale [38,39].

#### 2.2.1. Psychological Well-Being Scale

The Scale for Measuring Psychological Well-Being [2] contains 54 items, nine for each of the six subscales: autonomy, environmental mastery, personal growth, positive relationships with others, purpose in life, and self-acceptance. The scores for each of the sub-scales are calculated by summing the estimates of the nine corresponding items. The components of the psychological well-being construct identified by this scale were dependent variables in this study. The Scale for Measuring Psychological Well-Being is based on theoretical constructs of psychological well-being related to emotional health. The items should be evaluated on a Likert scale from 1 = strongly disagree, to 6 = strongly agree. A higher score means a higher level of psychological well-being. The overall Cronbach’s alpha coefficient for the Scale for Measuring Psychological Well-Being in the previous study was 0.84 [40]. Cronbach’s alpha coefficients in this study were as follows: autonomy 0.70, environmental mastery 0.63, personal growth 0.63, purpose in life 0.64, and self-acceptance 0.71 (data from the second survey).

#### 2.2.2. Emotion Assessment Questionnaire

The Multidimensional Emotion Questionnaire [4] includes two super-scales of emotional reactivity (positive and negative), scales that assess 10 discrete emotions (five positive and five negative), three subcomponents of positive and negative emotion (frequency, intensity, and persistence), and the regulation of both positive and negative emotions. Each of the 10 discrete emotions is evaluated according to four indicators: frequency, intensity, persistence, and regulation, each of the indicators on a 5-point scale. The scores for each of the 10 discrete emotions are calculated by summing scores of relevant frequency, intensity, and persistence items to form a single score for each of them. The scores of positive emotions’ frequency, intensity, and persistence are calculated by summing all relevant positive emotion scores. The scores of negative emotions’ frequency, intensity, and persistence are calculated by summing all relevant negative emotion scores. The scores of positive emotions overall and negative emotions overall are calculated by summing all relevant positive or negative emotions scores. Scores of regulation scales are calculated for positive emotion regulation and negative emotion regulation by summing scores for the relevant items. The components of the emotion construct identified by this scale were independent variables in this study. The items should be evaluated on a scale of 1 to 5. For example, happy: 1. How Often? 1 = About once per month or less; 5 = More than 3 times each day. 2. How Intense? 1 = Very Low; 5 = Very High. 3. How Long-Lasting? 1 = Less than 1 min; 5 = Longer than 4 h. 4. How Easy to Regulate? 1 = Very Easy; 5 = Very Difficult. The average estimate of discrete emotions is calculated by summing the partial estimates of that emotion expression: How Often? How Intense? How Long-Lasting?

Cronbach’s alpha coefficient was from 0.61 to 0.85 [4]. Cronbach’s alpha coefficients in this study were as follows: happy 0.70, sad 0.64, afraid 0.63, excited 0.66, angry 0.72, ashamed 0.62, enthusiastic 0.64, proud 0.71, anxious 0.66, and inspired 0.74 (data from second survey).

#### 2.2.3. Coping Measure Scale

The COPE 60-item instrument with 4 items per scale for a study of coping techniques and strategies has been developed [38], which was frustrating for subjects [39]. Therefore, a Brief COPE 28-items, 14-scale instrument with two items per scale was developed [39]. The scores of active coping, use of informational support, positive reframing, planning, emotional support, venting, humor, acceptance, religion, self-blame, self-distraction, denial, and substance use of the components of the coping construct are calculated by summing the two estimates of the respective items. The scores of components problem-focused coping, emotion-focused coping, and avoidance coping are calculated by summing the estimates of the relevant items. The components of the coping construct identified by this scale were independent variables in this study. Scales were singled out and verified by the developer, and Cronbach’s alpha coefficients ranged from 0.50 to 0.90 [39]. Scales can be divided into three higher levels as follows: super-scales problem-focused coping, emotion-focused coping, and avoidance coping [41]. The participants evaluate the statements on the scale in a four-point system from 1—“I haven’t been doing this at all” to 4—“I’ve been doing this a lot”. Cronbach’s alpha coefficients in this study were as follows: active coping 0.61, using instrumental support 0.68, positive reframing 0.61, planning 0.67, using emotional support 0.69, venting 0.67, humor 0.72, acceptance 0.62, religion 0.74, self-blame 0.63, self-distraction 0.68, denial 0.63, substance use 0.75, and behavioral disengagement 0.64.

### 2.3. Data Analysis

Data were analyzed using IBM SPSS for Windows 22.0. The values of skewness and kurtosis of all study variables ranged from 0.734 to −1.005, so, according to [42], if the variables are in the range from 2 to −2, the distribution of all variables does not significantly differ from the normal distribution and Student’s t criterion can be used for comparisons between means of scores of components of emotions, coping orientation, and psychological well-being constructs. The Wilcoxon Z test was used to compare the means between the first and second surveys because the samples are not independent, because they are from the same population of master’s students.

Descriptive statistics and Cronbach’s alpha coefficients for scales were calculated. The correlation between components of emotions, coping strategies, and psychological well-being constructs was calculated. The statistically significant level was set at *p* < 0.05.

## 3. Results

We highlight the results of the study after the second wave. After the first wave, there was a lot of stress on the public, but after the cancellation of the lockdown, there was much hope that the coronavirus disease would end. After the second wave, although the lockdown was canceled, there was a lot of information about new strains of coronavirus, and predictions about the end of the pandemic were not clear. Therefore, it was important to reveal how the master’s students’ main emotional indicators changed, what coping strategies were used, and how their other psychological well-being changed.

The processed data obtained from the survey using the Multidimensional Emotion Questionnaire of the components of the emotions construct are presented in Table 1 and Table 2.

Negative emotions were more pronounced after the first wave of COVID-19. Assessing the results of scores, the decreasing order of positive discrete emotions was: happy, excited, proud, inspired, and enthusiastic, and the order of negative discrete emotions was: sad, angry, ashamed, afraid, and anxious (Table 1). Positive emotions’ frequency, intensity, and persistence were, after the second wave, statistically significantly (*p* = 0.000) higher than those after the first wave, and vice versa, negative emotions’ frequency, intensity, and persistence were, after the second wave, statistically significantly (*p* = 0.000) less pronounced than after the first wave. However, after the second wave, negative persistence was rated higher than positive persistence, with scores of 15.20 and 13.15, respectively, so negative emotions were rated as prolonged. Positive emotions were rated better. Thus, the positive overall component was rated higher than the negative overall after the second wave.

Scholars sometimes evaluate only negative and sometimes only positive emotions. Estimates of the difficulties felt in regulating both positive and negative emotions were calculated separately and presented in Table 2 for convenience of analysis. The subjects had a much harder time regulating negative emotions than positive ones.

According to estimates of the regulation difficulties for positive emotion (Table 2), the decreasing order of emotion estimates was as follows: excited, inspired, enthusiastic, happy, and proud. Negative emotions according to the difficulty of their regulation can be arranged in descending order: sad, afraid, angry, anxious, and ashamed. After the first wave, the subjects found that it was more difficult for them to regulate both positive and negative emotions than after the second wave. The severity of regulation after the first wave was rated statistically significantly (*p* = 0.000) by higher scores for most discrete emotions, except for excited, whose scores differed not significantly.

Negative overall emotion regulation scores of 13.81 ± 2.29 were significantly higher than positive overall emotion regulation scores of 12.27 ± 2.44, (*p* = 0.000), which suggests that subjects rated negative emotions as more difficult to regulate than positives emotions (Table 2) after the second wave of coronavirus disease. The interesting result is that after the first wave, it was more difficult to regulate positive emotions than negative ones. This was probably due to hopes that the pandemic was over.

The active coping strategy was rated with the highest score after the second wave. The other six out of 14 coping strategies were ranked in descending order using instrumental support, venting, planning, humor, emotional support, and acceptance (Table 3). The least frequently used coping strategies were, in ascending order: denial, substance use, self-distraction, positive reframing, religion, self-blame, and acceptance. Problem-focused coping was the top rated, and avoidance-focused coping was the worst rated. The more often the coping strategy is used, the higher their rating.

The subjects indicated after the first wave that they mostly used a planning coping strategy. Other strategies used were positive reframing, self-blame, humor, acceptance, venting, and substance use. The evaluations of the active coping, use of instrumental support, planning, self-distraction, denial, substance use, and avoidance-focused coping strategies used differed significantly (*p* = 0.000) after the first and second COVID-19 waves. The subjects reported a significant reduction in substance use after the second wave, indicating that master’s students used problem-focused coping and emotion-focused coping strategies significantly more frequently during the second wave than during the first wave.

The psychological well-being of master’s students after the second wave was statistically significantly (*p* < 0.05) better than after the first wave in many indicators (Table 4). Environment mastery skills and personal growth desires did not change significantly, because many factors, such as lockdown conditions and university-determined study conditions, could not be managed by master’s students.

The component of the psychological well-being construct of the positive relations was rated the highest score (3.66) and the purpose in life the lowest score (3.25) after the second wave. Master’s students after the first wave rated highest (3.61) the purpose in life and lowest (3.21) the estimate of the autonomy.

The results of the study reveal that the restrictions imposed during the COVID-19 pandemic, the decline in social contacts, often isolation, and forced changes in study and living conditions intensified the expression of negative emotions and worsened master’s students’ psychological well-being.

The components of the psychological well-being construct Pearson correlation coefficients with the components of the emotion construct and with the coping strategy construct were calculated (Table 5). Due to the relatively small number of subjects (*n* = 128), the obtained values of correlation coefficients showed weak correlations in most of them, but some of them were statistically significant. Only statistically significant correlation coefficients are presented (Table 5).

Psychological well-being construct components had statistically significant relationships with emotions construct components: environment mastery with ashamed (−0.254) and self-acceptance with ashamed (−0.181).

The psychological well-being construct components had statistically significant relationships with coping strategy construct components: environment mastery with emotional support (0.193), acceptance (0.199), positive relation with self-blame (−0.261), self-acceptance with self-blame (−0.230), purpose in life with self-distraction (0.175), and positive relation with behavioral disengagement (−0.192).

Overall psychological well-being had statistically significant relationships with planning (−0.184), humor (−0.195), self-blame (−0.273), and self-distraction (0.177).

## 4. Discussion

This study aimed to reveal the expression of emotions, coping strategies, and psychological well-being of master’s students during the two waves of the COVID-19 pandemic period and the associations of emotion, coping, and psychological well-being.

Psychological well-being levels after the second wave of COVID-19 were significantly higher than after the first wave. Assessing the individual components, it was observed that the estimates of personal growth and environment mastery changed insignificantly. Estimates of autonomy, positive relation, purpose in life, and self-acceptance increased significantly. Students felt more frequent, intense, and lasting positive emotions.

Positives and negative emotions were easier for them to regulate than after the first wave of coronavirus disease. 

Students mostly used active coping, venting, and planning coping strategies to combat the stressful environment. These results are different from those obtained by other researchers. Thus, dominant coping strategies among students were acceptance, planning, and seeking emotional support. The least frequent strategies were substance use, denial, behavioral disengagement, and religious coping [35]. Active coping, use of instrumental support, and venting were the most frequently used, and the least frequently used were denial, substance use, and self-distraction coping strategies for master’s students, as found in our study. Self-acceptance, planning, and emotional support were more frequently used, and substance use, denial, behavioral disengagement, and religious coping strategies were the least frequently used, which were revealed in the study [35]. Two common least frequently used strategies for denial and substance use have been revealed both in our study and in a study of undergraduate students [35]. The study during and after the first lockdown revealed that during the first lockdown, overall psychological vulnerabilities increased and decreased during re-opening, but not to baseline, respectively. Subjects with increased vulnerability decreased their adaptive coping during lockdown [43]. The results of our study, however, after two lockdowns, showed an improvement in the level of psychological well-being, but a direct comparison of the results would be incorrect because the studies were conducted in different countries and samples. The use of instrumental support, planning, and denial coping strategies was found to be effective during the pandemic, while behavioral disengagement and self-blame strategies were found to be least effective [44]. The denial coping strategy could be associated with possible depression [44]. The denial coping strategy was used the least frequently among our study subjects; thus, cases of depression are unlikely, although we did not perform a specific study on the manifestations of depression. This is, in our point of view, an interesting fact, as, in Spain, for example, around 41% were detected as probable cases of depression [45]. Students with high levels of positive emotions used more active coping strategies, likely due to higher psychological capital [33]. Previous studies have already identified close links between emotions and the coping response of society [46]. High-risk perceptions of COVID-19 may likely help students transform negative emotions into positive ones, and they may feel more positive than negative emotions [33]. This is confirmed by the results obtained in our study, because after the second wave, as with most components of the psychological well-being construct, the overall estimates are higher than after the first wave of COVID-19.

Consistent with the results of our study, the application of master’s student problem-focused and emotion-focused coping strategies did not change significantly. However, the application of the avoidance-focused strategy declined significantly.

Adaptive coping strategies such as active coping, use of instrumental support, acceptance of emotional support, positive reframing, religion, and humor, classified as adaptive strategies, are strongly associated with psychological well-being [47,48]. Active coping, instrumental support, and acceptance strategies have been shown to have positive relationships with psychological well-being [49]. The emotion-focused strategy of self-blame was used relatively frequently in our study (4.94 points). Self-blame is reported to reflect responsibility for negative outcomes, and this is an internal, stable attribution to a stressor and is difficult to reverse [50]. Among avoidance strategies, subjects of our study most commonly used the behavioral disengagement strategy (4.48 points), which is used when individuals expect negative results [38]. Higher levels of psychological stress are known to be associated with avoidance coping strategies [51]. Students who most commonly used problem-focused coping strategies were better able to adapt to a changing environment and experience less stress, scholars have noted in previous studies [52].

The choice of coping strategy is significantly influenced by the stressful environment, and it has been revealed that significantly more nurses use a problem-focused strategy than nursing students [53]. Emotions of anger, fear, and anxiety were found to be significantly associated with problem-focused and emotion-focused strategies, but sadness was not associated with these strategies [53]. Scholars have revealed that the effectiveness of coping strategies is significantly related to the context of the situation. In situations with controlled factors, a problem-focused strategy is better suited, and an emotion-focused strategy is better suited as an environmental factor cannot be controlled [54]. Coping strategies that can be adjusted and adopted to stressful situations by undergraduates may positively affect their psychological well-being, as revealed in a previous study [55].

Possibly due to a disrupted, changed educational process, distance-learning students feel more negative emotions than the general population indicates [56]. College students have initiated more substance use as a coping tool against the psychological effects of the COVID-19 pandemic by combating negative emotions [57]. It should be noted that our master’s students surveyed gave a high grade to the substance use coping strategy after the first wave, which decreased significantly (*p* = 0.000) after the second wave.

A study of students’ coping strategies during the pandemic to combat emotional change found that students used problem-focused and emotion-focused significantly more often than avoiding coping strategies. Commonly used strategies included acceptance, emotional support, planning, and positive reframing. Among the least commonly used strategies were denial, substance use, and behavioral disengagement, thus classified as avoidance strategies [28].

The results of our study showed a significant increase (*p* = 0.000) in the happy, enthusiastic, and inspired scores of the components during the pandemic period from the end of the first wave to the end of the second wave, suggesting an increased level of adaptation to stressful environments. Meanwhile, sad, afraid, angry, ashamed, and anxious estimates decreased significantly (*p* = 0.000). Thus, after the first wave, students were more exposed to negative emotions than after the second wave. Likely, they have already at least partially adapted to the stressful environment, although they have been exposed to new strains of coronavirus. Positive emotions were already more frequent, more intense, and more persistent, and the expression of negative emotions weakened. Positive and negative emotions have become easier to regulate. Positive overall scores were higher and negative overall scores were lower than after the first COVID-19 wave. This, in our view, reflects an improvement in the adaptation skills to the stressful environment of master’s students.

The results of measurements conducted after the second wave of COVID-19 revealed that master’s students significantly (*p* < 0.05) were more likely to use problem-focused coping strategies. The venting and acceptance strategies were used significantly more often (*p* < 0.05) than the emotion-focused strategies. There was a significant decrease in avoidance-focused-type strategies such as self-distraction and denial (*p* < 0.05). However, there was a significant increase in substance use, although no specific explanation was provided in this study. The use of avoidance-focused strategies was significantly (*p* < 0.05) reduced. Students were likely to adapt to changing environments and study conditions and use problem-focused and emotion-focused coping strategies more often.

Scholars revealed that conscious involvement in activities that are significant to the individual can have a positive effect on psychological well-being [58]. Coping strategies that can be adjusted and adopted to stressful situations by undergraduates may positively affect their psychological well-being, as revealed in a previous study [59]. Problem-focused coping strategies were stronger than avoidance coping strategies associated with psychological well-being, as shown in the study [17]. Psychological well-being is significantly negatively affected by avoidance coping-type strategies for behavioral disengagement and venting, and self-blame [28]. We found that the self-blame strategy negatively correlates with the psychological well-being components of self-acceptance and positive relations. Links between the intensity of coping strategies and psychological well-being were revealed. The higher the psychological well-being level, the more intensive the application of coping strategies, and conversely, the lower the psychological well-being, the less intensive the application of coping strategies [23]. Coping strategies related to psychological support can help individuals combat the restriction of certain freedoms for greater well-being in society during a pandemic [60]. Our subjects reported significantly more problem-focused coping strategies after the second wave of coronavirus disease than after the first wave, and their psychological well-being rates were significantly higher than after the first wave. Thus, the results of our and other researchers confirm the importance of choosing coping strategies.

The results of our study show a significant increase (*p* < 0.05) in the happy, enthusiastic, and inspired scores of the components during the pandemic period from the end of the first wave to the end of the second wave, suggesting an increased level of adaptation to stressful environments. Meanwhile, sad, afraid, angry, ashamed, and anxious estimates decreased significantly (*p* < 0.05). Thus, after the first wave, students were more exposed to negative emotions than after the second wave. Likely, they have already at least partially adapted to the stressful environment, although they have been exposed to new strains of coronavirus. Positive emotions were already more frequent, more intense, and more persistent, and the expression of negative emotions were weakened. Positive and negative emotions have become easier to regulate. Positive overall scores were higher and negative overall scores were lower than after the first COVID-19 wave. This, in our view, reflects an improvement in the adaptation skills to the stressful environment of master’s students.

The score of the autonomy indicator increased significantly (*p* < 0.05) after the second wave, more social contacts could probably be had after the quarantine conditions were significantly relaxed, and some of the activities in university were not held using distance learning. This may have contributed to a significant increase in the positive relation score (*p* < 0.05). Interestingly, there was a significant decrease (*p* < 0.05) in one of the most important indicators of psychological well-being constructs: purpose in life. This may have been caused by altered social conditions, reduced communication opportunities, and study difficulties due to the use of distance learning. Overall psychological well-being also increased statistically significantly (*p* < 0.05).

The results of our study reveal an improvement in students’ ability to adapt to a stressful environment and this is, in our view, the consequence of the right choice of coping strategies. Scholars point out that adapted coping strategies have a positive effect on psychological well-being [55]. This is consistent with the observation by other scholars that the effectiveness of coping strategies is related to the context of the situation [54].

Thus, we can say that our hypothesis was partially confirmed. Some results confirm the statements made in the hypothesis, and some do not confirm them, as already noted above. Some of the results do not contradict the observations of other scholars, but some reveal other aspects, as with those in other studies.

### 4.1. Limitations of This Study

The study was not conducted with a representative sample. Therefore, to generalize the results of the study to the entire student population in Lithuania, all the more so, the results must not to be apply to all adults. The sample was not representative even of master’s students nationwide. Although the study took place after the lockdown’s cancellations and some restrictions, people’s fears persisted, and in addition, people were unaware of the rise of a new wave of coronavirus disease. There were also limitations in the fact that the data were collected in a self-report manner. No data were available on the level of master’s students’ psychological well-being before the COVID-19 pandemic, which complicated assessments of recovery rates. The strength of the study was that the survey was conducted twice and the continuity of the phenomena could be explained.

### 4.2. Future Research

Future research will cover larger representative samples, at least from a student perspective. In assessing the prevalence of depressive symptoms in society, the study will also include an examination of the level of depressive manifestations. The balance between study and leisure time will be explored to better understand changes in emotions, coping strategies, and psychological well-being.

Another pressing issue for future research is the coronavirus disease impact of the stressful environment for faculty staff and students due to a lockdown and the transition to online study. The results of studies already carried out on this topic show a significant impact on the working conditions and workload of professors [61]. However, there are other results. Thus, scholars, studying the impact of the transition from face-to-face classes to online learning, found that teachers’ satisfaction with their online learning experience was perceived as beneficial, and student’s satisfaction with their online learning experience was largely influenced by their perceived effectiveness of online learning technologies. However, the sudden shift from offline to online studying due to the COVID-19 pandemic has negatively affected students’ attitudes toward online studying [62].

## 5. Conclusions

The level of psychological well-being of students improved significantly after the second wave compared to the level after the first wave. Psychological well-being components, autonomy, positive relations, purpose in life, and self-acceptance ratings, also increased significantly.

The most commonly used coping strategies changed through a comparison of results after the first and second waves of coronavirus disease. Avoidance coping strategies were used significantly less frequently. Active coping and instrumental support strategies were used significantly frequently.

The overall level of positive emotions was significantly higher, and the negative ones were lower after the second wave of COVID-19 than after the first wave. Positive and negative emotions became easier to regulate, and students felt happier, less sad, and angrier after the second wave. Students felt positive emotions more often and more intensely.

The components of the psychological well-being construct were significantly correlated with the components of the emotions and coping constructs, indicating the importance of coping strategy selection and emotions management skills.

## Figures and Tables

**Table 1 ijerph-19-06014-t001:** Estimations of components of emotion after the first and second waves.

Component of Emotion Construct	Second Wave *n* = 128		First Wave *n* = 118		Wilcoxon Z	*p*
	M	SD	M	SD		
Happy	9.30	1.93	5.88	1.37	−9.314	0.000
Sad	8.95	2.27	12.18	1.38	−9.176	0.000
Afraid	8.36	1.87	10.20	2.24	−6.324	0.000
Excited	8.91	2.05	9.09	2.03	−0.368	0.071
Angry	8.69	2.08	10.70	1.82	−6.463	0.000
Ashamed	8.58	2.25	10.54	1.97	−6.171	0.000
Enthusiastic	8.41	2.32	6.16	1.37	−7.366	0.000
Proud	8.71	2.08	8.35	1.60	−1.108	0.268
Anxious	8.22	2.36	10.63	1.77	−7.075	0.000
Inspired	8.60	2.47	5.98	1.53	−7.837	0.000
Positive emotions frequency	16.14	2.89	11.92	2.24	−8.696	0.000
Positive emotions intensity	14.65	2.92	12.07	2.40	−6,698	0.000
Positive emotions persistence	13.15	2.82	10.18	1.81	−7.474	0.000
Negative emotions frequency	14.38	3.80	17.74	2.54	−8,067	0.000
Negative emotions intensity	13.45	3.10	18.22	2.28	−8,898	0.000
Negative emotions persistence	15.20	3.16	18.30	2.61	−6,962	0.000
Positive emotions overall	43.94	5.50	34.16	3.83	−9.914	0.000
Negative emotions overall	42.80	4.85	54.25	4.28	−9.652	0.000

**Table 2 ijerph-19-06014-t002:** Data on emotions regulation difficulties after the first and second wave.

Emotion	Second Wave *n* = 128		First Wave (*n* = 74)		Wilcoxson Z	*p*
	M	SD	M	SD		
Positive emotions						
Happy	2.14	0.820	3.95	0.772	−9.355	0.000
Excited	3.20	1.28	2.82	1.69	−1.66	0.097
Enthusiastic	2.34	0.835	3.89	0.844	−8.865	0.000
Proud	2.04	0.789	3.55	1.056	−8.484	0.000
Inspired	2.56	1.033	3.13	1.438	−3.518	0.000
Positive overall emotions regulation	12.27	2.44	18.13	2.38	−9.603	0.000
Negative emotions						
Sad	3.31	1.13	3.62	1.151	−2.019	0.044
Afraid	2.84	1.05	3.41	1.083	−4.206	0.000
Angry	2.56	0.802	2.99	0.856	−4.035	0.000
Ashamed	2.48	0.887	2.96	0.846	−4.314	0.000
Anxious	2.63	0.905	3.32	1.049	−5.240	0.000
Negative overall emotions regulation	13.81	2.29	16.36	2.11	−7.149	0.000

**Table 3 ijerph-19-06014-t003:** Estimations of components of coping after the first and second wave.

Component of Coping Construct	Second Wave *n* = 128		First Wave (*n* = 118)		Wilcoxon Z	*p*
	M	SD	M	SD		
Active coping	5.90	1.20	4.89	1.46	−4.329	0.000
Use instrumental support	5.65	1.41	4.86	1.70	−3.495	0.000
Positive reframing	4.83	1.66	5.24	1.87	−1.364	0.102
Planning	5.37	1.40	7.00	1.73	−6.152	0.000
Use emotional support	5.02	1.63	5.01	1.57	−0.050	0.960
Venting	5.44	1.39	4.95	1.96	−2.099	0.036
Humor	5.06	1.49	5.10	1.60	0.610	0.642
Acceptance	4.95	1.67	5.03	1.61	−0.283	0.777
Religion	4.90	1.56	4.64	1.56	−1.568	0.117
Self-blame	4.94	1.77	5.12	1.45	−0.491	0.623
Self-distraction	4.02	1.09	4.92	1.38	−4.973	0.000
Denial	3.54	0.99	4.68	1.61	−5.841	0.000
Substance use	3.91	0.96	4.98	1.68	−5.090	0.000
Behavioral disengagement	4.48	1.40	4.61	1.23	1.045	0.296
Problem-focused coping	2.72	0.337	2.77	0.662	−0.412	0.681
Emotion-focused coping	2.53	0.322	2.49	0.283	−0.820	0.412
Avoidance-focused coping	1.99	0.296	2.40	0.358	−7.293	0.000

**Table 4 ijerph-19-06014-t004:** Estimations of components of psychological well-being after the first and second wave.

Component of Psychological Well-Being Construct	Second Wave *n* = 128		First Wave (*n* = 118)		Wilcoxon Z	*p*
	M	SD	M	SD		
Autonomy	3.48	0.579	3.21	0.557	−3.466	0.001
Environmental mastery	3.48	0.484	3.38	0.438	−1.477	0.140
Personal growth	3.59	0.540	3.46	0.572	−1.267	0.205
Positive relations	3.66	0.464	3.29	0.511	−5.313	0.000
Purpose in life	3.25	0.645	3.61	0.592	−4.228	0.000
Self-acceptance	3.54	0.576	3.30	0.609	−2.938	0.003
Psychological well-being	3.50	0.217	3.37	0.267	−3.848	0.000

**Table 5 ijerph-19-06014-t005:** The correlation coefficients between psychological well-being and components of emotion and coping strategy constructs.

Component of Construct		Autonomy	Environment Mastery	Personal Growth	Positive Relations	Purpose in Life	Self-Acceptance
Emotion-Ashamed		0.069	−0.254 **	0.080	0.074	0.108	−0.181 *
Coping strategy	Emotional support	0.025	0.193 *	0.086	0.004	−0.080	0.061
	Self-blame	−0.161	−0.122	0.014	−0.261 **	0.027	−0.230 **
	Self-distraction	0.037	−0.017	−0.057	0.085	0.175 *	0.127
	Behavior disengagement	0.018	0.083	0.093	−0.192 *	−00.017	0.010

Notes: * *p* < 0.05 (two-tailed); ** *p* < 0.01 (two-tailed).

## Data Availability

The datasets collected and analyzed during the current study are available from the corresponding author on reasonable request. All survey data are password-protected.

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
