# Peer review of "The Emotions, Coping, and Psychological Well-Being in Time of COVID-19: Case of Master’s Students"

_ijerph, 2022, doi:10.3390/ijerph19106014_

Round 1

Reviewer 1 Report

The study is interesting and raises good questions about the pandemic. The methodology is ok, but not representative, as already mentioned in the text. It is not entirely clear why certain comparative studies wereselected and taken into account. That is, what added value does the study from Germany provide? In some places there are errors in punctuation,grammar and syntax. Please check the text again.

Author Response

Dear reviewer,

the authors thank to the reviewer for their work and valuable comments.

Kind regards

Audrone Dumciene

Reviewer 2 Report

I thank the authors for their interesting work. The subject matter is interesting, and the instruments and methods used are appropriate and adequately justified. However, there are some aspects that should be clarified before publication of the manuscript:

  1. In the section on Materials and Methods, the authors explain that the target population of the study consists of Lithuanian students at the master's level. The results of the study are strongly conditioned by the nationality of the students. However, this has not been indicated in the definition of the problem, in the title or in the formulation of the hypothesis.
  2. It is not clear to me why the authors have chosen master's students instead of, for example, university students in general. Perhaps the reason is to be found in the specific characteristics of the educational system of the country where the study is framed, but this should be clarified by the authors.
  3. In section 2.3, the authors refer to the "variables" of the study to analyze the internal consistency of the scales analyzed. However, these variables have not been defined previously.
  4. I sense that the area of knowledge of the participating students significantly conditions the perceptions expressed (or, at least, many of them). Have the authors explored the possibility of considering the area of knowledge as an independent variable of the study? I suggest employing the Kruskal-Wallis test to analyze gaps by knowledge area in the students' responses.
  5. It would be useful to explain to what extent the specific policies carried out by the Lithuanian health authorities regarding the pandemic have conditioned the perceptions expressed by the students.
  6. I suggest briefly discussing the results obtained in relation to other analogous studies but focused on university faculty, rather than students. In this regard, I suggest including the following references:

https://doi.org/10.3390/ijerph19063732

https://doi.org/10.1080/10963758.2021.1907196

  1. I recommend that the authors make a thorough linguistic revision of the text. For example, line 60 reads "The mental health and well-being of adults was", where it should read "were". There are numerous textual errors of this style in the manuscript.

Author Response

Dear reviewer,

the authors thank you for their work and valuable comments.

Kind regards

Audrone Dumciene

Reviewer 3 Report

Comments on “The emotions, coping and psychological well-being in time of COVID-19:

case of master’s students”

  1. Outline of the paper

The purpose of this paper is to uncover changes in emotions, coping strategies, and psychological well-being of master students during a pandemic. The psychological well-being of master’s students after the second wave was better than after the first wave. Significant associations were reveled between the same components of psychological well-being, emotion and coping strategies. This study showed that the master's students improved their adaptive abilities probably in the environment of long-term exposure to coronavirus disease, as most psychological well-being indicators were improved significantly after the second wave

  1. Major comments

I found that the paper is well written scientifically. It is quite important to show the empirical evidence on the coping strategies during pandemic. We need more discussion on the relation between the policy measure for pandemic and the changes in coping strategies. It is quite important to evaluate the policy measures taken during the pandemic with observing the behavioral changes, especially coping strategy to the stresses. In addition, we need more information on the vulnerability of students, especially the type of psychological weakness. This kind of information would be useful for designing the policy measures for caring people with distresses during the pandemic.

  1. Minor comments

In Table 5, “Copin” should be “Coping”.

Author Response

(The authors gave the same response as above.)

Round 2

Reviewer 2 Report

The authors have responded in a reasoned and detailed manner to all the questions and comments I have asked them. As a result, the manuscript has, in my opinion, been improved. I thank the authors for their work and congratulate them on it.